# Tracking clonal and plasmid transmission in colistin- and carbapenem-resistant *Klebsiella pneumoniae*

Ifeoluwa Akintayo,[1] Marko Siroglavic,[2] Daria Frolova,[3] Mabel Budia Silva,[1] Hajo Grundmann,[1] Zamin Iqbal,[3,4] Ana Budimir,[2] Sandra Reuter[1]

**ABSTRACT** The surveillance of mobile genetic elements facilitating the spread of antimicrobial resistance genes has been challenging. Here, we tracked both clonal and plasmid transmission in colistin- and carbapenem-resistant *Klebsiella pneumoniae* using short- and long-read sequencing technologies. We observed three clonal transmissions, all containing Incompatibility group (Inc) L plasmids and New Delhi metallo-beta-lactamase $bla_{NDM-1}$, although not co-located on the same plasmid. One IncL-$bla_{NDM-1}$ plasmid had been transferred between sequence type (ST) 392 and ST15, and the promiscuous IncL-$bla_{OXA-48}$ plasmid was likely shared between a singleton and a clonal transmission of ST392. Plasmids within clonal outbreaks and between clusters and STs had 0–2 single nucleotide polymorphism (SNP) differences, showing high stability upon transfer to same or different STs. The simplest explanation, without a comprehensive analysis with long-read sequencing, would be the spread of a single common IncL-$bla_{NDM-1}$ plasmid. However, here, we report $bla_{NDM-1}$ in five different plasmids, emphasizing the need to investigate plasmid-mediated transmission for effective containment of outbreaks.

**IMPORTANCE** Antimicrobial resistance occupies a central stage in global public health emergencies. Recently, efforts to track the genetic elements that facilitate the spread of resistance genes in plasmids outbreaks, utilizing short-read sequencing technologies, have been described. However, incomplete plasmid reconstruction from short-read sequencing data hinders full knowledge about plasmid structure, which makes the exploration very challenging. In this study, we used both short- and long-read sequencing in clinical *Klebsiella pneumoniae* from University Hospital Centre Zagreb, Croatia, which was resistant to both last-resort antibiotics colistin and carbapenem. Our results show complex transmission networks and sharing of plasmids, emphasizing multiple transmissions of plasmids harboring carbapenem and/or colistin resistance genes between and within *K. pneumoniae* clones. Only full-length sequencing plus a novel way of determining plasmid clusters resulted in the complete picture, showing how future active monitoring of plasmids as a vital tool for infection prevention and control could be implemented.

**KEYWORDS** *Klebsiella pneumoniae*, plasmid analysis, whole-genome sequencing, transmission, carbapenem resistance, colistin resistance

Antimicrobial resistance (AMR) is one of the biggest challenges in modern medicine and a major concern to public health worldwide as reported by the World Health Organization (https://www.who.int/publications/i/item/9789241509763) (1). According to the European Centre for Disease Prevention and Control (ECDC) annual report of the European Antimicrobial Resistance Surveillance Network (EARS-Net), there are high and increasing antibiotic resistance among Gram-negative bacteria in many parts of

Address correspondence to Sandra Reuter, Sandra.reuter@uniklinik-freiburg.de.

S.R. declares travel, accommodation, and speaker's honoraria from Illumina, Ltd.

See the funding table on p. 10.

Europe, including Croatia (2), one of which is *Klebsiella pneumoniae*—a major cause of hospital-acquired infections.

In recent years, carbapenem-resistant *K. pneumoniae* (CRKP) lineages have become pathogens of major concern, especially in the healthcare system (3), owing to their ability to acquire resistance to last-resort antimicrobials, like carbapenems and colistin, thereby making infections particularly difficult to treat, according to the Centers for Disease Control and Prevention (CDC) (https://doi.org/10.15620/cdc:82532) (4). Resistance to carbapenems can be mediated by mutations in genes that lead to overexpression of efflux pumps or to porin deficiency, along with expression of ambler class C (AmpC) beta-lactamases or extended spectrum beta-lactamases (ESBL). However, the most worrying is the resistance due to acquisition of genes encoding carbapenem-hydrolyzing enzymes (i.e., carbapenemases) (5). The major carbapenemases implicated in resistance to common carbapenems among Enterobacterales are $bla_{KPC}$, $bla_{OXA-48}$, $bla_{NDM-1}$, and $bla_{VIM}$ (6). Therapeutics deployed for the treatment of multidrug and CRKP infections involve the use of antibiotics of last resort, such as colistin (polymyxin E) (https://www.ecdc.europa.eu/en/publications-data/carbapenem-resistant-enterobacteriaceae-second-update) (7). As with carbapenem resistance, colistin resistance (ColR) is on the rise worldwide and has been reported in several parts of Europe, including Croatia (8). Mechanisms of bacterial resistance to colistin involve lipopolysaccharide modifications, mediated by mutations on chromosomal genes (*mgrB*, *phoPQ*, and *pmrAB*), and acquisition of mobile colistin resistance (*mcr*) genes, which have been reported in *K. pneumoniae* (9). Until now, different variants of the mobile colistin resistance gene *mcr* (*mcr*-1 to *mcr*-10) have been identified in several bacterial species from various sources (10). The rapid dissemination of carbapenem resistance genes is most often driven by mobile genetic elements such as plasmids, which are transferable within and between bacterial species or strains (11). Outbreaks that are plasmid associated are increasingly studied and have been reported to facilitate the spread of antimicrobial resistance genes in multiple bacterial strains or species, especially in clinical settings (12, 13). Hence, outbreak investigations of carbapenem- and colistin-resistant strains and their mobile elements are critical to understand epidemiology and efficiently strategize infection prevention and control measures. In this study, we investigated an outbreak of colistin- and carbapenem-resistant *K. pneumoniae* (ColR-CRKP) strains in a university hospital in Zagreb, Croatia. Short- and long-read sequences were used to investigate and differentiate clonal and plasmid-mediated transmission of antimicrobial resistance.

## RESULTS

### Basic overview of the population

We included 46 isolates ColR-CRKP from the University Hospital Centre (UHC) Zagreb, isolated over a nearly 2-year period. We identified two major sequence types (STs), ST15 ($n = 29$) and ST392 ($n = 8$), with other isolates belonging to ST101 ($n = 4$), ST17 ($n = 1$), ST147 ($n = 1$), ST273 ($n = 1$), ST274 ($n = 1$), and ST5081 ($n = 1$) (Table S1). Genetic information based on single nucleotide polymorphism (SNP) distance did not indicate a clonal transmission cluster in ST101. The metadata showed that the patients were not at any point in time in the same ward; hence, we focused our clonal transmission analysis on ST392 and ST15.

### Clonal transmission patterns of ST392 and ST15

We determined genetic relatedness of ST392 isolates (Fig. 1A). The phylogeny showed two distinct clusters (Fig. 1A) that present putative separate transmission chains separated by 14–22 SNPs. There was a singleton isolate Z002 distinct from all the others by 7–8 SNPs. Cluster I involved two patients (Z047 and Z013) with 0 SNP apart; the patients overlapped in a surgical intensive care unit (ICU) ward in February 2022. The isolates were non-susceptible to all antibiotics tested except to fosfomycin (Table S1). A

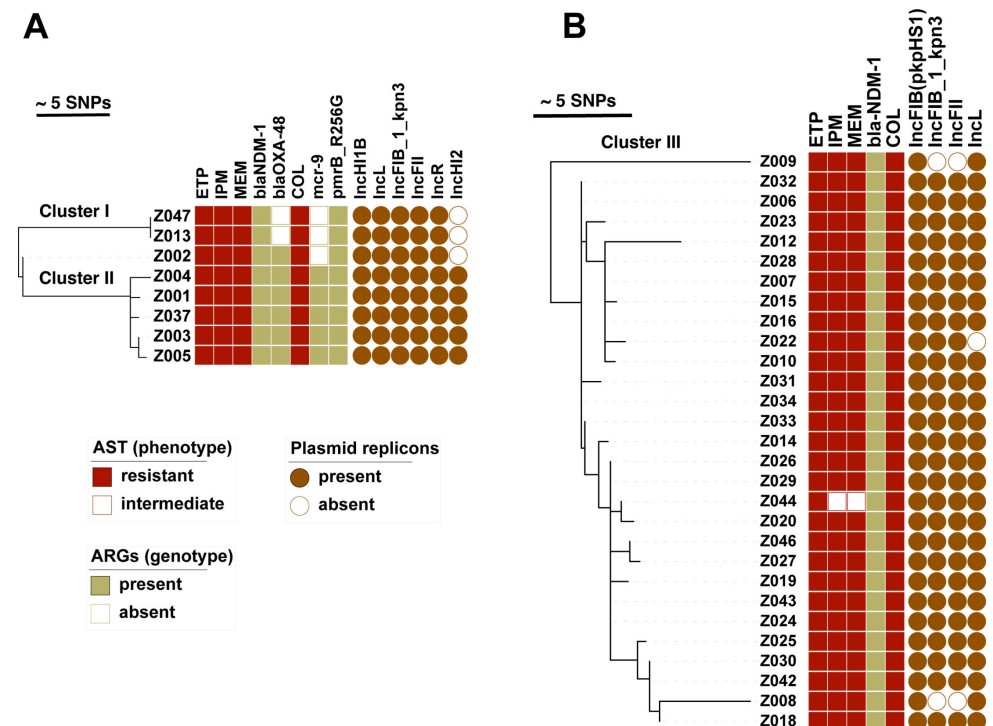

**FIG 1** Phylogenetic reconstruction of main STs. Phylogeny of (A) ST392 and (B) ST15. Results of phenotypic testing for carbapenem and colistin resistance are followed by their corresponding genotype. Distribution of main detected plasmid incompatibility groups is shown. ETP, ertapenem; IPM, imipenem; MEM, meropenem; COL, colistin.

mutation in the *pmrB* gene (*pmrB*_R256G) was the only colistin resistance determinant detected in cluster I. Cluster II involved five patients with 1–4 SNPs apart, with patients overlapping in an internal medicine ICU ward between September and December 2021. Isolate Z004 was temporally the first isolate detected, before other patients entered the internal medicine ICU, and can therefore be considered the index patient of cluster II. The isolate is highly non-susceptible and showed phenotypic pandrug resistance to nine classes of antibiotics, including polymyxins and carbapenems (Table S1). The other isolates in the cluster were non-susceptible except to fosfomycin (Table S1). Cluster II had both *pmrB*_R256G and *mcr*-9. Carbapenemase *bla*_NDM-1 was detected in all ST392 isolates, with cluster II and Z002 additionally carrying *bla*_OXA-48. A full set of detected resistance genes can be found in Table S1. A total of five plasmid replicons were detected, including IncHI1B, IncFII, IncFIB, IncR, and IncL, with an additional plasmid replicon, IncHI2, detected only in cluster I (Fig. 1A).

Looking at ST15 (Fig. 1B and 2), we identified 29 patients in seven different wards over 6 months, thus the phylogeny showed more genetic diversity. Except for isolate Z009, which was 15–22 SNPs apart, the other isolates had 0–11 SNP difference, mean and median of 3 SNPs. This ST15 cluster we referred to as cluster III. There were two main wards involved in this outbreak, a COVID-19 (coronavirus disease 2019) ward and a neurology ward. A number of patients only had temporal overlaps but no direct ward contact with other patients of this cluster (Z012, Z030, Z027, Z043, Z046, Z044; Fig. 2); however, their position within the genetic diversity within this outbreak suggested they were part of it (Fig. 1B). As with ST392, ST15 was also highly non-susceptible with phenotypic resistance toward six classes of antibiotics (Fig. 1B; Table S1). The only carbapenemase detected was *bla*_NDM-1. No known colistin resistance determinant was detected among the isolates (Table S1). Four different plasmid replicons were found, IncFIB(pkpHS1), IncFIB_1_kpn3, IncFII, and IncL (Fig. 1B). Spatial and temporal relation-

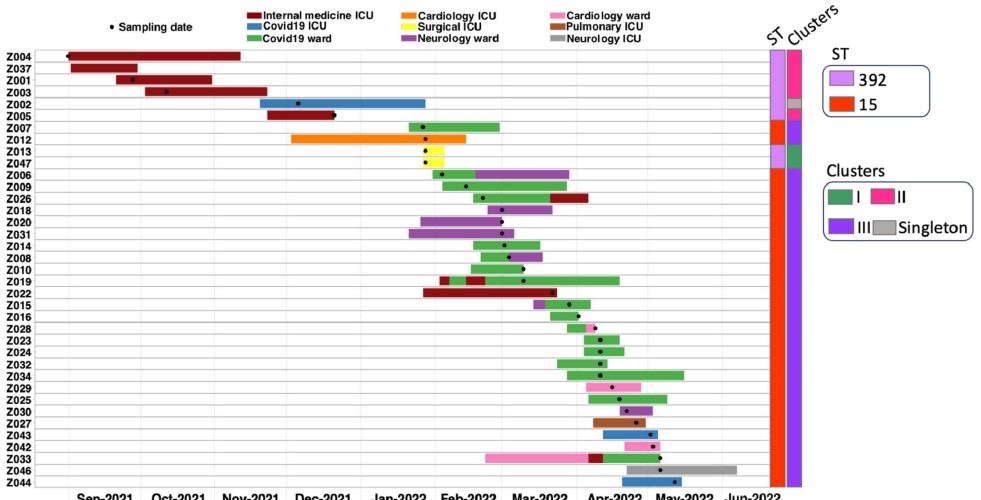

FIG 2 Spatial and temporal relationship of the patients within the hospital for the two main STs and clonal transmission clusters. Patients are sorted by the isolation date of their ColR-CRKP, with the stay on a ward depicted by different colors. Identified ST and cluster allocation is given.

ship of the patients involved in each clusters showed that they were at some point in the same ward at the same time (Fig. 2).

## Plasmid makeup and transmission of IncL-type plasmids

The variation in the genotypic resistance pattern and plasmid types in this collection prompted further investigation into the carbapenemases and the plasmids carrying them. We detected common replicon types and common resistance genes, namely IncL plasmids and $bla_{NDM-1}$; however, the exact makeup of these plasmids was unknown, as was the location of $mcr$-9.

The IncL plasmid replicon was seen in ST15 excluding Z022, all isolates of ST392 and ST5081. From comparison analysis based on mash distances (Fig. 3A), clade 1 comprised IncL plasmids found in ST392-cluster II and ST392 singleton Z002. Clade 2 comprised IncL plasmids found in isolates of ST392-cluster I, ST15 (cluster III), and ST5081. The plasmids in clade 1 had a size of 64 kb and carried only the $bla_{OXA-48}$ gene; hence, we referred to it as IncL$_{-64kb}$. The plasmid sequence was compared to the public database using Basic Local Alignment Search Tool (BLAST) and was found to be the promiscuous, well-documented pOXA-48 plasmid. This plasmid is highly conserved, with little variation in size and few mutations, within the ST392-cluster II and the singleton isolate. The IncL plasmid in clade 2 had a size of ~96 kb and carried the $bla_{NDM-1}$ gene with other genes that confer resistance to aminoglycosides, chloramphenicol, sulfonamides, and quinolones. In this study, we referred to this plasmid as IncL$_{-96kb}$. When compared to the public database using BLAST, we found only partial query coverage (68% and 64% to accessions OW970501 and OW969621, respectively) corresponding to the typical pOXA-48 IncL plasmid regions but with a deletion of $bla_{OXA-48}$. Both IncL$_{-64kb}$ and IncL$_{-96kb}$ plasmids thus shared a common backbone including the conjugational transfer system, with differences found in the AMR regions (Fig. 3B; Fig. S2). The novel IncL$_{-96kb}$ plasmid showed 0–2 SNPs within clusters and upon the transfer between STs, with only an insertion of an additional $insB$ insertion element and hypothetical protein in ST15 and 5081, respectively. The occurrence of such a distinct and unique novel IncL$_{-96kb}$ plasmid and high degree of conservation suggested that it was spreading between the ST392-cluster I, ST15-cluster III, and ST5081-singleton within the hospital.

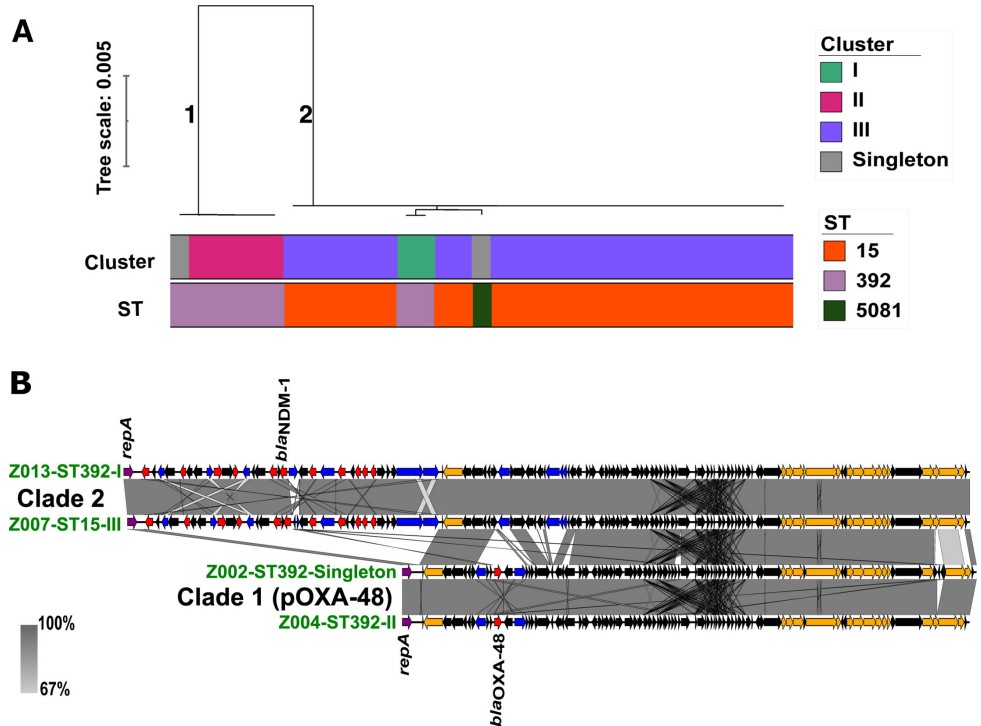

**FIG 3** Comparative genomics analysis of IncL plasmid. (A) Mashtree showing IncL plasmid clustering. Clade 1 is IncL$_{64kb}$ from ST392-cluster II and singleton Z002 while clade 2 is IncL$_{96kb}$ in ST15-cluster III, ST392-cluster I, and ST5081-singleton. (B) Linear map showing the genetic content of the plasmids from two representatives from each clade, clade 1 (IncL$_{64kb}$ pOXA-48) and clade 2 (IncL$_{96kb}$). The gray area indicates regions of shared similarities. Red arrows indicate resistance genes, blue arrows indicate transposons, orange arrows indicate the conjugal transfer genes, and black arrows are either hypothetical proteins or genes of the replication system.

## Characterization of the mobile genetic context of *bla*$_{NDM-1}$ and *mcr*-9

ST392-cluster II isolates contained an IncHI2 *bla*$_{NDM-1}$/*mcr*-9 plasmid with sizes ranging from 276 to 281 kb (Table S2). Other genes conferring resistance to aminoglycosides (*aac*-(3)-*IIg*, *aac*-(6)-*IIc*), beta-lactam (*bla*$_{OXA-1}$, *bla*$_{TEM-1}$), chloramphenicol (*catB3*), sulfonamides (*sul1*), trimethoprim (*dfrA*-19), and quinolones (*qnrB1*) were also carried. The IncHI2 plasmids were highly similar, having a 96%–100% coverage and a 99% identity with each other. After mapping Illumina short reads of all ST392 cluster II isolates against the hybrid assembly of the longest plasmid, the plasmids were 0–2 SNPs apart with mash distances ranging from 1.4e−05 to 1.4e−03. From the comparative analyses, we observed two small regions differing in isolate Z005 accounting for the observed size difference (Fig. S3B through D), the first of which was potentially by insertion sequence (IS)-mediated excision because it was flanked by repeats. This patient was the last sampled, with the isolate collected 2.5 months after the previous sample. The plasmid was compared to the public database using BLAST and was found to be closely related to an *mcr*-9 harboring IncHI2 plasmid (accession no. CP030742.1) found in another *K. pneumoniae* isolate, although this plasmid did not contain *bla*$_{NDM-1}$.

In contrast, the *mcr*-9 and *bla*$_{NDM-1}$ genes detected in singleton ST274 (Z036) were carried on separate plasmids, IncHI2 and IncC, respectively. Comparative structural analysis of the IncHI2 plasmids using pling revealed that five previously described plasmids in ST392 cluster II are almost identical structurally; however, the other IncHI2 plasmid (from Z036) is considerably different, separated by at least 12 structural events (Fig. S3A). The pling output is a relatedness network where nodes are plasmids, and edges are labeled with two numbers: a "containment distance" (what proportion of the smaller plasmid is not alignable to the larger) and a rearrangement distance (how

many rearrangements/indels separate these two plasmids) (14). The region carrying the $bla_{NDM-1}$ gene is absent. Aside from that, the Z036 IncHI2 plasmid had a large inversion with further genes lost (Fig. S3B).

The IncC plasmid (175 kb) carrying the $bla_{NDM-1}$ gene in Z036 had similarities (Mash distance 1.4e−03) with the IncC plasmid (163 kb) carrying $bla_{NDM-1}$ found in Z011 (ST101 singleton), except for a region lost in Z011 containing $qnrA6$ bounded by insertion sequence (IS) element and hypothetical proteins (Fig. S4). When compared to the public database, we found identical IncC $bla_{NDM-1}$ plasmids in *Escherichia coli* and *K. pneumoniae* (100% query coverage and percentage identity; MG450360.1, CP030744.1).

## Multiple plasmids carrying carbapenemases in other STs

The remaining isolates all had one or two carbapenem resistance gene(s) carried on a variety of plasmids (Table S2). $bla_{OXA-48}$ was seen in all ST101 isolates (Z011, Z021, Z038, and Z039) and was carried on an IncR-FIA plasmid. Apart from Z038, the other IncR-FIA plasmids clustered together, having a 100% shared sequence content but are separated by 5–7 rearrangement events (Fig. S5). When all the IncR-FIA plasmids were compared to the public database using BLAST, we found a 99%–100% query coverage and percentage identity to IncR-FIA plasmids (CP083022.1 and MN218814.1) carrying $bla_{OXA-48}$ found in *K. pneumoniae* isolated in Switzerland and Serbia, respectively, in the year 2017.

Z045 (ST147) carried a $bla_{NDM-1}$ gene on an IncFIB. Z035 (ST17) and Z017 (ST273) had a $bla_{VIM-1}$ gene carried on an IncN plasmid; however, we did not recover this plasmid in Z017 (ST273) in the long-read sequencing data; it may have been lost in culture. Z017 short reads were mapped against the Z035 hybrid assembled and showed that the Z017 IncN plasmid is 0–2 SNPs different from Z035 (ST17), thus a possible plasmid transfer may have occurred between these two isolates. When compared to the database using BLAST, the Z035 IncN plasmid showed only a 93% query coverage and a 99.6% identity to a single plasmid (CP070567.1) without $bla_{VIM-1}$ found in a clinical *K. pneumoniae*.

Cluster analysis for plasmids without carbapenemases can be found in Figure S6.

## DISCUSSION

Our study investigated ColR-CRKP isolates from UHC Zagreb. Although we detected common carbapenemases as well as shared plasmid Inc types, when looking in more detail using long-read sequencing, we found a complex picture of clonal transmission as well as plasmid exchange (summarized in Fig. 4), involving the dissemination of $bla_{NDM-1}$, $bla_{OXA-48}$, and $mcr$-9 resistance genes via clonal and horizontal transfer.

From a standard outbreak investigation, we detected three clonal transmissions—two within ST392 (cluster I and cluster II) and one in ST15 (cluster III) (Fig. 1). ST392 has been sporadically identified from hospital-acquired infections, either as a carbapenem resistant or a susceptible clone in different parts of Europe, including Spain, Netherlands, United Kingdom, Italy, France, Germany, Luxembourg, Belgium, Türkiye, and Austria (15), while ST15 is known to be one of the dominant CRKP lineages of *K. pneumoniae* in clinical samples from European countries, including Croatia, alongside ST11, ST101, and ST258/512 (15). These lineages are considered "high risk" clones that have gained a foothold in most parts of southern and Eastern Europe, and are most often associated with hospital outbreaks. The earliest report of carbapenemases in Croatia were $bla_{NDM-1}$ and $bla_{KPC}$ in 2008 and 2012, respectively, detected in *K. pneumoniae* isolates from UHC Zagreb patients (16, 17).

From a plasmid perspective, we identified two major IncL plasmids—the promiscuous pOXA-48 (IncL$_{-64kb}$) shared between ST392-cluster II and ST392-singleton, and a novel 96 kb plasmid (IncL$_{-96kb}$) shared between ST392-cluster I, ST15-cluster III, and ST5081-singleton (Fig. 3 and 4). The IncL/M plasmid is known for its worldwide dissemination of $bla_{OXA-48}$, and its emergence with $bla_{NDM-1}$ therefore makes it a great public health concern (18). Recently, the IncL/M plasmid group has been re-classified into separate IncL and IncM plasmids, where $bla_{OXA-48}$ is carried on the IncL plasmid and $bla_{NDM-1}$ is carried on an IncM plasmid (19). In our study, however, we found an IncL pOXA-48

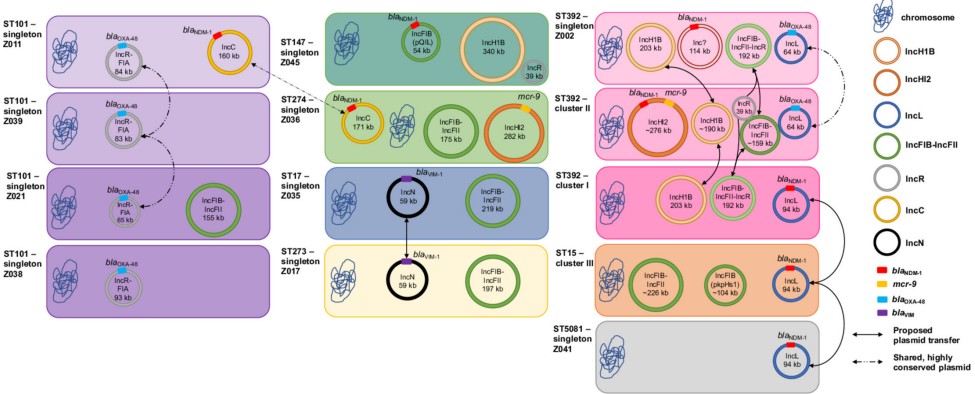

**FIG 4** Overview of proposed clonal transmission clusters and horizontal plasmid transfers. Bacterial STs and cluster, plasmid incompatibility groups are highlighted by different colors, as are the relevant antibiotic resistance genes. Proposed horizontal plasmid transfers are indicated by arrows, and inconclusive transfers by dashed lines.

backbone (IncL$_{-64kb}$) fused with an additional region that contained numerous AMR genes including $bla_{NDM-1}$ (Fig. S2). While mobile genetic elements, such as plasmids, harboring antibiotic resistance genes are known to undergo significant gene rearrangement (20), our study showed high stability of this novel IncL$_{-96kb}$ plasmid even when transferred within and between clusters and STs. Given the conserved nature of pOXA-48 and geographically widespread reports of it, we cannot distinguish between the uptake of this plasmid IncL$_{-64kb}$ from a common source or plasmid exchange in our two ST392 clusters. In contrast, given that the novel IncL$_{-96kb}$ has not been described elsewhere and it is highly conserved, this makes horizontal transfer locally between the different STs very likely. Other plasmids carrying $bla_{OXA-48}$ are less common but have been described, like the IncR-FIA (21) which we also found in this collection, as well as an IncFII plasmid (22). The IncN plasmid was also likely shared between two STs, but we were unable to recover it in the long-read sequencing. An IncC $bla_{NDM-1}$ plasmid was observed in two STs, but as we found identical plasmids in the public databases, we cannot ascertain whether it was truly shared or acquired independently. IncC has been reported to play an important role in the rapid dissemination of carbapenemases, with $bla_{NDM-1}$ commonly carried on this plasmid, and has been seen in several bacterial species including *Klebsiella* species as seen in our study (23).

From the perspective of the carbapenemases, we also found a complex pattern. Apart from the IncL$_{-96kb}$ plasmid described above, we found $bla_{NDM-1}$ on a 276 kb IncHI2 and on a 163 kb IncC (Table S2; Fig. 4). In IncHI2, the carbapenemase was found together with *mcr*-9 (Fig. S3). This plasmid replicon type has been identified as a carrier of multiple resistance genes, and also due to its size, it was termed a superplasmid. These plasmids are not limited to *K. pneumoniae* (24) and are reported to be key drivers for the spread of *mcr*-9 in Enterobacterales, including *K. pneumoniae* (25). $bla_{OXA-48}$ was found not only on the typical IncL$_{-64kb}$ pOXA-48 but also on several IncR-FIA-type plasmids.

In terms of resistance, all isolates collected as part of this study were highly resistant, and we detected a carbapenemase in all of the isolates (Table S1). Although all isolates were classed as phenotypically resistant to colistin, only eight isolates harbored a known mutation (*pmrB*_R256G) conferring resistance to colistin. This suggests that there are unknown chromosomal mutations contributing to colistin resistance (26). We detected *mcr*-9 in five ST392cluster II isolates as well as one ST274 isolate. However, the contribution of this gene toward colistin resistance is not clear (27–29). A previous study observed the co-localization of *mcr*-9 and carbapenemase genes ($bla_{NDM-1}$, $bla_{VIM-1}$, and $bla_{OXA-48}$) in 26 *K. pneumoniae* isolates belonging to ST274 and ST147 from patients in eight European countries (30), with ST147 being a single locus variant of ST392.

Overall, while we observed a good overlap of patients in the same wards at the same time, we do not have epidemiological support for all clonal transmissions and none for the proposed horizontal plasmid transfer. There might be intermediate patients who were not screened and where colonization might have gone undetected, and who might be able to close these transmission gaps. Furthermore, only few colonies or single colony picks were selected for each patient, thus colonization with more than one clone of a particular species might have gone undetected. Additionally, for this study, only carbapenem- and colistin-resistant *K. pneumoniae* strains were considered, missing out on colistin-susceptible *K. pneumoniae* as well as other Enterobacterales that might harbor the plasmids under investigation. Lastly, environmental samples were not considered either.

In conclusion, we provided conclusive evidence for clonal transmission of colistin- and carbapenem-resistant *K. pneumoniae* in Croatia, as well as for horizontal plasmid transfer between different STs. Mobile genetic elements have been a driving force in the dissemination of antimicrobial resistance genes most especially in healthcare-associated bacterial infections (14); however, tracing the transfer of plasmids has been challenging. Short-read sequencing offers limited resolution and would have mislead us in our current study setting: while we detected numerous isolates of the same ST, and both IncL and $bla_{NDM-1}$ as well as $bla_{OXA-48}$ and *mcr*-9, in a number of our isolates, the actual picture of spread and dissemination was more complex. Our study therefore emphasizes the importance of bacterial whole-genome surveillance with both short- and long-read sequencing as a critical tool for detecting clonal and horizontal transmission of extremely drug-resistant pathogens, and then devising appropriate infection prevention and control strategies accordingly.

## MATERIALS AND METHODS

### Culture, DNA extraction, and whole-genome sequencing

A total of 46 *K. pneumoniae* isolates from the University Hospital Centre (UHC) Zagreb, Zagreb, Croatia, were identified as part of routine surveillance and clinical testing of microbiological samples from all clinical departments. Clinical samples comprised 54% of tested samples. Samples were cultured on Columbia blood agar containing 5% sheep blood. Isolate identification was confirmed by MALDI-TOF mass spectrometry (Bruker Microflex LT, Bremen, Germany). Antimicrobial susceptibility was determined by VITEK 2 compact system (bioMérieux, Paris, France) and was interpreted according to the European Committee on Antimicrobial Susceptibility Testing (EUCAST) criteria (available at http://www.eucast.org/clinical_breakpoints/). For colistin MICs, this was performed by broth microdilution (BMD). Information on all samples can be found in Table S1.

These samples were then transferred to the Medical Center–University of Freiburg, Freiburg, Germany, for whole-genome sequencing (WGS). Species identification was confirmed using MALDI-TOF mass spectrometry (BRUKER), and isolates were cultured overnight at 37°C on blood agar. DNA was extracted according to the manufacturer's instructions using a high pure PCR template preparation kit (Roche Diagnostics). All isolates were short-read sequenced with an Illumina MiSeq using Nextera DNA flex library preparation V2 300 cycle PE kit according to the manufacturer's instructions, and all isolates were long-read sequenced with Oxford Nanopore Technology (ONT) GridION platform using the ligation protocol SQK-LSK109 with native barcode EXP-NBD104 kit and SQK-LSK 114 (Oxford Nanopore Technology, Oxford, UK) in accordance with the manufacturer's protocol. The sequencing was performed using FLO-MIN106 (v.R9.4) and FLO-MIN114 (v.R10.4.1) flow cell type.

### Quality control of sequence analysis

*De novo* assemblies of the Illumina sequence reads were generated using SPAdes v3.13.1 with kmer sizes 21, 33, 55, 77, 99, 109, and 123 (31). Assemblies were then filtered to

only include contigs with a minimum of 500 bp. The sequence type was determined using multi-locus sequence typing (MLST) v2.10 (T. Seemann, unpublished data; https://github.com/tseemann/mlst) (32). All sequence reads were mapped to the reference genome MGH78578/ATCC700721 (CP000647) using SMALT v0.7.6 (33), and the reads were checked for the respective species using kraken (34) with the minikraken 4 GB reference database. Quality control (QC) criteria include read allocation to the respective expected genus, coverage greater than 30×, expected genome size (5–6.5 Mbp), number of contigs less than 500, largest contig greater than 100,000 bp, N50 greater than 100,000 bp, and identification of MLST alleles (Table S1).

ONT fast5 generated output file was basecalled on the GridION using guppy v5.0.12. Sequence reads were trimmed using porechop v0.2.4 (https://github.com/rrwick/Porechop) (35) to remove adapters from read ends and within the reads, using default parameters. The trimmed reads were filtered to retain long reads using filtlong v0.2.1 (https://github.com/rrwick/filtlong) (36). Hybrid assemblies were generated using unicycler v0.5.0 (37), and contigs with less than 500 bp were excluded. QUAST v5.0.2 (38) was used to generate the assembly statistics.

All sequencing statistics as well as accession numbers for raw reads and assemblies can be found in Table S1. Sequencing data were deposited under ENA project PRJEB76496.

## Phylogenetic analysis and isolate characterization

SNPs were filtered from the mapping data with genome analysis toolkit (GATK) (39), the variant filtered files were converted to a fasta file, where SNP sites and absent sites (N) were replaced in the reference genome. An alignment file was generated, and mobile genetic elements were removed (https://github.com/sanger-pathogens/remove_blocks_from_aln). Gubbins was employed to detect recombination events and reconstruct a phylogenetic tree (40). Trees were visualized using figtree v1.4.4 (https://github.com/rambaut/figtree) and the interactive tree of life (iTOL) v6.5 (41). Antimicrobial resistance genes and point mutations were identified from the assembled files using AMRfinderplus and its NCBI database (42).

## Plasmid analysis and visualization

Plasmid replicons were predicted using ABRicate and PlasmidFinder database (43). The plasmid contigs were extracted from the hybrid assembled files and annotated using bakta (44). Plasmids were clustered in two independent ways. First, they were clustered using MASH v1.4.5 (45), using a Mash distance of threshold of 0.001 to define highly similar plasmid sequences as previously described (46). For the structural relatedness of the plasmid, pling v1.0.1 with default parameters was used (47). Similarity searches were performed using NCBI BLAST (48), linear comparative analyses of the plasmid were generated using Easyfig v2.2.2 (49), and circular plot of the novel plasmid was generated using DNAplotter (50). To check the SNP distance between the plasmids, the Illumina short reads of the isolates were mapped to the full hybrid assembly of the longest plasmid. An alignment file was generated, all regions except the plasmid were masked, SNP sites were generated, and SNP distance was calculated.

## ACKNOWLEDGMENTS

We would like to thank Leonardo Duarte dos Santos for excellent technical assistance.

S.R. and I.A. are supported by the German Ministry of Education and Research (BMBF) through grant 01KI2018 to S.R. as a junior research group leader.

S.R. and A.B. conceived the study. I.A., D.F., M.B.S., and S.R. analyzed the data. M.S., H.G., Z.I., and A.B. contributed resources. I.A., M.S., A.B., and S.R. wrote the manuscript.

## AUTHOR AFFILIATIONS

[1]Institute for Infection Prevention and Control, Faculty of Medicine, University of Freiburg, Freiburg, Germany

[2]Department of Clinical Microbiology, Infection Prevention and Control, University Hospital Centre Zagreb, Zagreb, Croatia

[3]European Molecular Biology Laboratory–European Bioinformatics Institute, Hinxton, United Kingdom

[4]Milner Centre for Evolution, University of Bath, Bath, United Kingdom

## AUTHOR ORCIDs

Ifeoluwa Akintayo  http://orcid.org/0000-0002-4234-7649
Sandra Reuter  http://orcid.org/0000-0003-1672-5789

## FUNDING

| Funder | Grant(s) | Author(s) |
| --- | --- | --- |
| Bundesministerium für Bildung und Forschung (BMBF) | 01KI2018 | Sandra Reuter |

## AUTHOR CONTRIBUTIONS

Ifeoluwa Akintayo, Data curation, Formal analysis, Methodology, Visualization, Writing – original draft, Writing – review and editing | Marko Siroglavic, Data curation, Methodology, Writing – original draft, Writing – review and editing | Daria Frolova, Formal analysis, Methodology, Visualization, Writing – review and editing | Mabel Budia Silva, Formal analysis, Methodology, Writing – review and editing | Hajo Grundmann, Resources, Writing – review and editing | Zamin Iqbal, Methodology, Resources, Visualization, Writing – review and editing | Ana Budimir, Conceptualization, Resources, Writing – original draft, Writing – review and editing | Sandra Reuter, Conceptualization, Formal analysis, Funding acquisition, Investigation, Methodology, Resources, Supervision, Visualization, Writing – original draft, Writing – review and editing

## DATA AVAILABILITY

Sequencing data were deposited under ENA project PRJEB76496.

## ADDITIONAL FILES

The following material is available online.

### Supplemental Material

**Fig. S1 (mSystems01128-24-s0001.pdf).** Spatial and temporal relationship of all patients within the hospital.
**Fig. S2 (mSystems01128-24-s0002.pdf).** Circular plasmid map of the novel IncL-96kb plasmid.
**Fig. S3 (mSystems01128-24-s0003.pdf).** Structural and mutational analysis of IncHI2 plasmids.
**Fig. S4 (mSystems01128-24-s0004.pdf).** Comparative genomic analysis of IncC plasmid.
**Fig. S5 (mSystems01128-24-s0005.pdf).** Pling relatedness network of IncR-FIA plasmids shows differences in gene content and organization.
**Fig. S6 (mSystems01128-24-s0006.pdf).** Structural comparative genomics of the IncHI1B and IncFIB-FII-R plasmids not carrying carbapenemases.
**Legends (mSystems01128-24-s0007.docx).** Legends for supplemental material.
**Table S1 (mSystems01128-24-s0008.xlsx).** Metadata of collection.

**Table S2 (mSystems01128-24-s0009.xlsx).** Overview of all detected plasmids, their incompatibility groups, sizes, and colistin and/or carbapenem resistance (COL-CR) genes detected.

## Open Peer Review

**PEER REVIEW HISTORY (review-history.pdf).** An accounting of the reviewer comments and feedback.

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
