## [Reviewer comments · mSystems]

TRACKING CLONAL AND PLASMID TRANSMISSION IN COLISTIN AND CARBAPENEM RESISTANT *KLEBSIELLA PNEUMONIAE*

Ifeoluwa Akintayo, Marko Siroglavic, Daria Frolova, Mabel Silva, Hajo Grundmann, Zamin Iqbal, Ana Budmir, and Sandra Reuter

Corresponding Author(s): Sandra Reuter, Medical Center University of Freiburg

Review Timeline:

Submission Date:	August 19, 2024
Editorial Decision:	October 11, 2024
Revision Received:	November 11, 2024
Accepted:	December 2, 2024

Editor: Angela Re

Reviewer(s): The reviewers have opted to remain anonymous.

Transaction Report:

DOI: <https://doi.org/10.1128/msystems.01128-24>

Re: mSystems01128-24 (TRACKING CLONAL AND PLASMID TRANSMISSION IN COLISTIN AND CARBAPENEM RESISTANT *KLEBSIELLA PNEUMONIAE*)

Dear Dr. Sandra Reuter:

Revision Guidelines

Sincerely,
Angela Re
Editor
mSystems

Editor (Comments for the Author)

Authors are invited to be careful about the usage of abbreviations without any explanation through all the manuscript, including the abstract.

Panels B) and C) in Figure 3 do not convey any message. Please, authors are invited to remove them..

Reviewer #1 (Comments for the Author):

Regarding the manuscript "Tracking clonal and Plasmid transmission in colistin and carbapenem resistant *Klebsiella pneumoniae*".

The authors investigate a collection of 46 *K. pneumoniae* isolates obtained from screening and clinical testing, from UHC Zagreb in Croatia. The isolates were grouped by sequence type, and clonal relationships were investigated together with epidemiological data. Further, plasmids were recovered by hybrid assemblies, and plasmid spread/outbreaks were investigated.

Overall, the manuscript is contemporary and relevant. Understanding the impact of plasmids as a driver for AMR in clinical settings is crucial.

I have some minor and specific comments, that from my view will improve the manuscript.

- The authors are mixing present and past tense throughout the manuscript.
- Some sentences are quite long, and hard to follow.
- In the result section, the authors jump from describing one cluster to another, and then back to the first. In the figure legends, the same is happening. I would recommend that they finish up describing one cluster, and then turn to the next. Same scenario with the figure legends, describe one sub-figure, and then turn to the next. The constant mixing makes it a little difficult to follow.
- Additionally, part of the result section should go into the discussion, as you are discussing the results. Or you could consider to rewrite and combine results and discussion into one section. This is a rather short paper, so it could benefit from combining the sections.
- Could be an idea to include a small table in the result section, summarizing the features of the different clusters. Eg. name, number of isolates, STs, resistance genes, phenotype.
- Not all gene and species names are italicized.

Line 24-27: I would rephrase this sentence, as you do not mention a hypothesis that you reject. Eg:

"The simplest explanation, without a comprehensive analysis with long read sequencing, would be the spread of a single common IncL-blaNDM-1 plasmid. However, here we report blaNDM-1 found in five different plasmids, emphasizing the need to investigate plasmid-mediated transmission for effective containment of outbreaks."

Line 30-33: I would also rephrase this sentence to eg.:

"Recently, efforts to track the genetic elements that facilitate the spread of resistance genes in plasmid outbreaks, utilizing short read sequencing technologies, have been described. However, plasmid reconstruction from short read sequencing hinders full knowledge about plasmid structure, which makes the exploration very challenging."

Line 90: Did you perform MALDI-TOF twice?

Line 108: You mention expected genome size, but not the expected number.

Line 120: Is it the mapping data from line 105-106? Meaning that the MGH78578 is reference for Kraken as well as your SNP analysis?

Line 135-137: "However this approach..." this is more of a discussion.

Line 138-141: "The output is a relatedness..." this belongs to results and not method.

Line 180: Introduce cluster III in the text.

Line 219: "The novel IncL-96kb plasmid showed 0-2 SNPs..." You are not describing SNPs on the plasmids in your method section. Which reference and method are you using? Please add that to the manuscript method section.

Line 232: "After mapping illumina short read..." I guess it is the same as mentioned above?

It makes sense that you are using the illumina short reads for the SNP/mapping due to the error rate.

Line 249-253: Some parts of this are more discussion like and not results.

Line 25-261: Belongs to discussion

Line 296: Bracket is missing around reference 36

Line 310: Closing bracket is missing after supp. Fig 2.

These figures get a little messy with the load of information. I would recommend to remove redundant sub-figures.

Figure 1: For me, figure 1a is redundant. Fig 1b and Fig 1c should be same size in relation to each other. Fig 1b is relatively big compared to 1c.

Figure 3: Fig 3a. could be redundant.

Figure 4: Resolution is low, quality gets bad when zooming in.

Reviewer #2 (Comments for the Author):

Akintayo and colleagues wrote a very relevant, very interesting, well-written, concise and easy to read paper about colistin and carbapenem-resistant *K. pneumoniae*. The work that has been done very thoroughly with state of the art analysis techniques

on a genetically diverse bacterial collection. However, clonal spread of CPE and plasmid transmission studies analyzed by short-read and long-read sequencing technologies are done on a large scale by countries all over the world and are published in lower impact journals. The use of MASH and PLING is a good addition in these clonal/plasmid transmission studies and has been presented with beautiful figures.

The following minor issues should be addressed or clarified:

- Broth microdilution (BMD) is best method to determine colistin resistance, and not Vitek. Can colistin AST be confirmed with BMD?
- *mcr-9* is not conferring resistance to colistin, although often falsely implicated in colistin resistance in many reports.
- gene names/bacterial species should be in italics, also in the reference list. Please change throughout.

Editor

Comments	Response
Authors are invited to be careful about the usage of abbreviations without any explanation through all the manuscript, including the abstract.	Thank you for the observation, this has been corrected.
Panels B) and C) in Figure 3 do not convey any message. Please, authors are invited to remove them..	These panels have now been removed from the figure.

Reviewer #1

Comments	Response
The authors are mixing present and past tense throughout the manuscript	Thank you for the observation. We have looked through the manuscript and to the best of our ability to correct all instances of improper use of tense.
Some sentences are quite long, and hard to follow.	We have gone through our manuscript again to shorten instance of long and hard sentences.
In the result section, the authors jump from describing one cluster to another, and then back to the first. In the figure legends, the same is happening. I would recommend that they finish up describing one cluster, and then turns to the next. Same scenario with the figure legends, describe one sub-figure, and then turn to the next. The constant mixing makes it a little difficult to follow.	Thank you, we have now described each cluster before moving to the next cluster, see lines 91-122.
Additionally, part of the result section should go into the discussion, as you are discussion the results. Or you could consider to rewrite and combine results and discussion into one section. This is a rather short paper, so it could benefit from combining the sections.	Thank you for your suggestions. We have tried to keep to presenting results with only discussion into inclusion into clusters and clades, before the overall discussion section.
Could be an idea to include a small table in the result section, summarizing the features of the different clusters. Eg. name, number of isolates, STs, resistance genes, phenotype.	Thank you for your recommendation. We agree that it is a good idea to show each cluster and its features and that is why we have this information in the supplementary table S1.
Not all gene and species names are italicized.	Thank you for your observation, this has now been corrected.
Line 24-27: I would rephrase this sentence, as you do not mention a hypothesis that you	This has been rephrased as suggested, see lines 25-28

reject. Eg: "The simplest explanation, without a comprehensive analysis with long read sequencing, would be the spread of a single common IncL-blaNDM-1 plasmid. However, here we report blaNDM-1 found in five different plasmids, emphasizing the need to investigate plasmid-mediated transmission for effective containment of outbreaks."	
Line 30-33: I would also rephrase this sentence to eg.: "Recently, efforts to track the genetic elements that facilitate the spread of resistance genes in plasmids outbreaks, utilizing short read sequencing technologies, have been described. However, plasmid reconstruction from short read sequencing hinders full knowledge about plasmid structure, which makes the exploration very challenging."	Thank you. This has been rephrased as suggested. Lines 30-34
Line 90: Did you perform MALDI-TOF twice?	Yes, MALDI-TOF was done at the UHC Zagreb and when it was sent to University of Freiburg for sequencing, we reidentified to rule out contamination.
Line 108: You mention expected genome size, but not the expected number.	Thank you for the observation, this number has now been added (Line 316). The expected size ranges from 5-6.5 Mbp which is the size that has been reported from other Klebsiella genomic studies.
Line 120: Is it the mapping data from line 105-106? Meaning that the MGH78578 is reference for Kraken as well as your SNP analysis?	Thank you for pointing this out. The statement was quite misleading and has now been rephrased. Mapping the reads to MGH78578 is not an input for kraken, the raw reads were used for taxonomic check with the kraken database. Lines 313-314
Line 135-137: "However this approach..." this is more of a discussion.	This has now been restructured. Lines 343-344
Line 138-141: "The output is a relatedness..." this belongs to results and not method.	This has now been added to result. Lines 171-175
Line 180: Introduce cluster III in the text.	Thank you, this has been done. Lines 111-114
Line 219: "The novel IncL-96kb plasmid showed 0-2 SNPs..." You are not describing SNPs on the plasmids in your method section. Which reference and method are you using? Please add that to the manuscript method section.	This has now been added. Lines 347-350
Line 232: "After mapping illumina short	Thank you for the comment. We have

read..." I guess it is the same as mentioned above? It make sense that you are using the illumina short reads for the SNP/mapping due to the error rate.	clarified this point now (Lines 347-350)
Line 249-253: Some parts of this is more discussion like and not results.	Thank you for pointing this out, it has now been removed from result
Line 25-261: Belongs to discussion	This was in both result and discussion. We have now removed it from the result section. Lines 239-240
Line 296: Bracket is missing around reference 36	Thank you for the observation, this has been corrected.
Line 310: Closing bracket is missing after supp. Fig 2.	This has been corrected, thank you for the observation. Lines 227
There figures get a little messy with the load of information. I would recommend to remove redundant sub-figures. Figure 1: For me, figure 1a is redundant. Fig 1b and Fig 1c should be same size in relation to each other. Fig 1b is relatively big compared to 1c.	Fig. 1A has been removed whereas Fig 1B and C have been fixed.
Figure 3: Fig 3a. could be redundant.	Thank you for the suggestion. We would like to show how related the plasmids are in relation to the STs and clusters. Most especially because 3b and c have now been removed.
Figure 4: Resolution is low, quality gets bad when zooming in.	Thank you for the observation, the figure has now been replaced with a better resolution

Reviewer #2

Comment	Response
Broth microdilution (BMD) is best method to determine colistin resistance, and not Vitek. Can colistin AST be confirmed with BMD?	Thank you for pointing this out, BMD was used for colistin. This has been added to the manuscript. Lines 294-295
mcr-9 is not conferring resistance to colistin, although often falsely implicated in colistin resistance in many reports.	We agree with reviewer that there are conflicting reports on the specific genes or mutations that confer resistance to colistin. While isolates carrying these genes are often phenotypically resistant, many of our colistin-resistant isolates lack known genotypic resistance determinants. We believe that a future comprehensive study is needed to identify the underlying mechanisms of colistin resistance, particularly with mcr-9 . However, this is beyond the scope of the present study. We have clarified this point in the paper (lines

	256-257)
gene names/bacterial species should be in italics, also in the reference list. Please change throughout.	Thank you for your observation

Re: mSystems01128-24R1 (TRACKING CLONAL AND PLASMID TRANSMISSION IN COLISTIN AND CARBAPENEM RESISTANT *KLEBSIELLA PNEUMONIAE*)

Dear Dr. Sandra Reuter:

Your manuscript has been accepted, and I am forwarding it to the ASM production staff for publication. Your paper will first be checked to make sure all elements meet the technical requirements. ASM staff will contact you if anything needs to be revised before copyediting and production can begin. Otherwise, you will be notified when your proofs are ready to be viewed.

Sincerely,
Angela Re
Editor
mSystems

Reviewer #1 (Comments for the Author):

I had the pleasure of reviewing both the original and revised versions. The authors have addressed all my questions and concerns to my satisfaction. I have no further comments.